# Post-translational modifications glycosylation and phosphorylation of the major hepatic plasma protein fetuin-A are associated with CNS inflammation in children

Frederik Ricken[1,2], Ahu Damla Can[1,2], Steffen Gräber[2], Martin Häusler[1☉], Willi Jahnen-Dechent[2☉] *

1 Division of Neuropediatrics and Social Pediatrics, Department of Pediatrics, RWTH Aachen University Hospital, Aachen, Germany, 2 Helmholtz Institute for Biomedical Engineering, Biointerface Laboratory, RWTH Aachen University Hospital, Aachen, Germany

☉ These authors contributed equally to this work.

* willi.jahnen@rwth-aachen.de

**Data Availability Statement:** Anonymised data are available as Supporting Information files.

## Abstract

Fetuin-A is a liver derived plasma protein showing highest serum concentrations in utero, preterm infants, and neonates. Fetuin-A is also present in cerebrospinal fluid (CSF). The origin of CSF fetuin-A, blood-derived via the blood-CSF barrier or synthesized intrathecally, is presently unclear. Fetuin-A prevents ectopic calcification by stabilizing calcium and phosphate as colloidal calciprotein particles mediating their transport and clearance. Thus, fetuin-A plays a suppressive role in inflammation. Fetuin-A is a negative acute-phase protein under investigation as a biomarker for multiple sclerosis (MS). Here we studied the association of pediatric inflammatory CNS diseases with fetuin-A glycosylation and phosphorylation. Paired blood and CSF samples from 66 children were included in the study. Concentration measurements were performed using a commercial human fetuin-A/AHSG ELISA. Of 60 pairs, 23 pairs were analyzed by SDS-PAGE following glycosidase digestion with PNGase-F and Sialidase-AU. Phosphorylation was analyzed in 43 pairs by Phos-Tag™ acrylamide electrophoresis following alkaline phosphatase digestion. Mean serum and CSF fetuin-A levels were 0.30 ± 0.06 mg/ml and 0.644 ± 0.55 µg/ml, respectively. This study showed that serum fetuin-A levels decreased in inflammation corroborating its role as a negative acute-phase protein. Blood-CSF barrier disruption was associated with elevated fetuin-A in CSF. A strong positive correlation was found between the CSF fetuin-A/serum fetuin-A quotient and the CSF albumin/serum albumin quotient, suggesting predominantly transport across the blood-CSF barrier rather than intrathecal fetuin-A synthesis. Sialidase digestion showed increased asialofetuin-A levels in serum and CSF samples from children with neuroinflammatory diseases. Desialylation enhanced hepatic fetuin-A clearance via the asialoglycoprotein receptor thus rapidly reducing serum levels during inflammation. Phosphorylation of fetuin-A was more abundant in serum samples than in CSF, suggesting that phosphorylation may regulate fetuin-A influx into the CNS. These results may help establish Fetuin-A as a potential biomarker for neuroinflammatory diseases.

**Funding:** This study was supported by the German Research Foundation (DFG, TRR 219, Project ID 322900939 and Project ID 403041552, both awarded to WJ-D) and by the START program of the Faculty of Medicine of the RWTH Aachen University (grant number 129/14, awarded to MH). The funders had no role in study design, data collection and analysis, decision to publish, or preparation of the manuscript.

**Competing interests:** The authors have declared that no competing interests exist.

## Introduction

Fetuin-A (alpha2-Heremans-Schmid glycoprotein, genetic symbols *AHSG/FETUA*) is a 60 kDa glycoprotein which is predominantly synthesized in liver cells, abundant in serum [1], attains highest serum concentrations in utero and in preterm children and reaches normal serum levels soon after birth [2]. Synthesis is repressed during acute phase by proinflammatory cytokines TNF-alpha, IL-1beta or IL-6, but is induced by high blood glucose and glucocorticoid levels as well as growth factors and insulin [3–5]. The cystatin-like amino-terminal protein folding domain CY1 mediates high affinity binding of fetuin-A to apatite in bone and teeth [6]. Fetuin-A is modified post-translationally in several ways. It is processed by limited proteolysis and carries up to six serine and one threonine FAM20C phosphorylation sites [7–9]. Sequence prediction and experimental work according to Uniprot entry P02765 list altogether two N-linked and six O-linked glycosylation sites [10].

Glycosylation is important for cell-protein interaction, regulating protein internalization, protein degradation, regulation of cell growth, differentiation and death [11, 12]. Glycans are involved in leucocyte-trafficking, i.e. adhesion of immune cells to endothelial cells [13]. Impaired glycosylation may lead to organ malformation and dysfunction and enhance tumor cell survival [11]. Fetuin-A has altogether eight putative glycosylation sites with terminal sialic acid. Of these, six O-glycans were shown to comprise core 1 or core 8 glycans, containing N-acetylgalactosamine, galactose and terminal sialic acids attached to T270, S280, S293, T339, T341 and S346, according to Uniprot [10, 14]. Two complex N-glycans containing N-acetylglucosamine, hexose and terminal sialic acids are attached to N156 and N176, respectively [10, 14].

Sialic acid monosaccharides decorate both O- and N-glycans. Posttranslational acetylation or hydroxylation further increase variability. Sialic acids are cleaved by sialidases located on the cell surface, in the cytoplasm and in microorganisms [15]. Increased sialidase activity was observed in septicemia [16]. Sialic acids may stabilize molecules and membranes, interact with cells and the extracellular matrix, protect proteins and glycan-bound monosaccharides from proteases, glycosidases and from oxidative stress. Sialic acids induce immune tolerance, inhibit immune cells and regulate immune cell migration [15, 17]. The terminally bound sialic acids are of particular importance for fetuin-A metabolism as they protect circulating fetuin-A from hepatic clearance through the asialoglycoprotein receptor, which is expressed on hepatocytes and liver macrophages [18, 19]. Attenuated fetuin-A sialylation has been observed in autoimmune diseases [17], in neonates with intrauterine growth retardation [20] and in patients with rheumatoid arthritis [21], whereas increased sialylated fetuin-A serum levels were recorded in patients with allergies [22].

Most studies on fetuin-A glycosylation indicate a major role of fetuin-A sialylation for immune regulation. This includes counter regulation of the innate immune response in a rat model of carrageenan injection [23], augmentation of the phagocytosis of apoptotic cells by human peripheral blood monocyte-derived macrophages [24], promotion of Th2 immune response in mice [22], a neuroprotective effect in a rat model of cerebral stroke [25] and a protective effect on survival in an endotoxin-model of septicemia in mice [26]. Sialylated fetuin-A is thought to be a carrier of spermine, which inhibits secretion of proinflammatory cytokines from macrophages and monocytes [27].

Protein phosphorylation regulates enzyme and receptor activity modulating metabolic activity and protein-protein interactions with downstream signaling [28–30]. Several phosphorylated proteins have been discussed as biomarkers for neurological diseases, including transferrin [31], alpha synuclein [32] or phosphorylated neurofilament H [33], but only phosphorylated tau has so far been established as a biomarker of neurodegenerative disorders [34].

Fetuin-A plays an important role in mineralized matrix metabolism [35]. Fetuin-A inhibits precipitation of calcium phosphate from supersaturated salt solutions [36]. Phosphorylated fetuin-A was shown to preferentially bind calcium phosphate [37] forming complexes similar or identical to calciprotein particles (CPP), which we and others showed to stabilize calcium and phosphate thus preventing mineral precipitation and ectopic calcification [35, 38–40]. Fetuin-A acts as an inhibitor of dystrophic calcification together with other extracellular regulators of mineralization including magnesium and pyrophosphate [41]. Fetuin-A phosphorylation was also claimed critical for inhibiting insulin receptor binding and signaling [42], a role soon debated [43, 44] and not convincingly confirmed thereafter. The role of phosphofetuin-A in cerebrospinal fluid has never been studied.

In brain tissue strong fetuin-A staining has been reported for prenatal age suggesting its presence throughout brain development [45]. An intrathecal synthesis of fetuin-A in the choroid plexus was described in rats [46]. Although the amount of cerebral fetuin-A staining rapidly decreases after birth, fetuin-A presence was confirmed in pathological conditions including multiple sclerosis, ischemia and infections [47].

Altered protein composition of the cerebrospinal fluid (CSF) is of diagnostic value in various neurological diseases. In this context, basic diagnostic tools are assessment of the blood-CSF barrier and of intrathecal immunoglobulin synthesis as modelled by Reiber et al. [48]. Accordingly, proteins gain access to the CSF either by passive entry from intravascular fluid (80%), or else are synthesized intrathecally, particularly by the choroid plexus (20%). This results in a concentration gradient with higher protein concentrations in ventricular than in lumbar CSF [49]. The concentration of plasma-derived proteins in CSF depends on their concentration in serum, the concentration gradient between plasma and CSF and the flow rate of the CSF [50]. Slowing of the CSF flow rate in inflammatory processes leads to an increase of the CSF protein concentration [51]. Similarly, reduced CSF flow rates in the elderly and in neonates may contribute to their higher CSF protein concentrations [50]. Hereby the blood-CSF barrier is considered a functional barrier. Disturbed blood-CSF barrier function with high CSF protein concentrations and increased CSF/serum protein quotients may also be based on a slow CSF flow rate. Calculation of the age-dependent CSF albumin/serum albumin quotient (QAlb) is used for clinical assessment of the blood-CSF barrier function. Accordingly, pathological intrathecal immunoglobulin synthesis is diagnosed based on CSF and serum quotients for antibodies (QIg) and albumin (QAlb) [49, 50].

In addition to blood-CSF barrier function and intrathecal immunoglobulin synthesis various CSF proteins have been screened in order to establish biomarkers that may help diagnose certain neurological diseases or monitor their activity [52]. As for multiple sclerosis these candidate biomarkers include transcription factors, neurofilaments, receptors, microRNA, antibodies to viruses and also fetuin-A [52, 53]. Fetuin-A levels in CSF correlated with disease activity [54] and fetuin-A knock-out mice were protected from experimental autoimmune encephalitis (EAE) [55]. In line with its role as a negative acute phase protein, fetuin-A serum levels were reported to decline with inflammatory activity [56]. Cognitive decline in the elderly was associated with both low fetuin-A serum and CSF concentrations [57, 58].

The aim of this study was to gain knowledge on the physiology of posttranslational fetuin-A modifications, i.e. of glycosylation and phosphorylation, in the CSF compartment as this might help to assess its suitability as a biomarker for inflammatory CNS disease. We determined amount, glycosylation, and phosphorylation of fetuin-A in blood and CSF of probands with and without cerebral inflammatory disease.

## Methods

### Probands

Paired Blood and CSF samples had been obtained from 66 children treated at the Childrens´ hospital of RWTH Aachen University Hospital during venous and lumbar punctures performed for routine clinical analysis. The study was approved by the Ethics Committee at the Medical Faculty of RWTH Aachen University (EK 139/07). Written parental consent (and patient consent so far applicable) was available for all probands. Long-term storage of the samples was done at -80˚ Celsius.

Clinical data collected included basic clinical diagnoses, age at investigation, serum C-reactive protein concentrations, CSF/serum quotients for IgG and albumin, CSF cells and protein levels and blood count. The laboratory tests were performed as part of routine clinical testing by the central laboratory of the RWTH Aachen University hospital. Total serum protein was measured using Biuret method (Total Protein Gen. 2 Kit; cobas c 701 module for clinical chemistry; Roche Diagnostics International AG, Rotkreuz, Switzerland). Total CSF protein was measured using turbidimetry (Total Protein Urine/CSF Gen. 3 kit; cobas c 501 module for clinical chemistry; Roche Diagnostics International AG, Rotkreuz, Switzerland). Albumin in CSF and serum was assessed using immunonephelometry (N Antiserum to Human Albumin; BN ProSpec nephelometer, Siemens Healthcare GmbH, Erlangen, Germany). IgG was measured in CSF and Serum using immunonephelometry (N Antiserum to Human-IgG, BN ProSpec nephelometer, Siemens Healthcare GmbH, Erlangen, Germany). CSF cells and blood count was assessed using flow cytometry.

For measurement of fetuin-A in serum and CSF using ELISA, paired blood and CSF samples of 66 probands were obtained including 27 samples from probands without an inflammatory CNS disease, 35 samples from patients with an inflammatory disease of the CNS and 4 samples without reliable classification in one of the two groups. Clinical data is shown in Table 1, diagnoses are displayed in Table 2.

Glycosylation was studied in 23 paired CSF/serum samples, including 10 controls and 13 samples derived from patients with inflammatory diseases of the CNS. Clinical data and diagnoses are shown in Tables 3 and 4.

For phosphorylation studies paired blood and CSF samples of 43 probands were investigated including 10 controls and 33 samples from patients with inflammatory CNS diseases. Clinical diagnoses and data are shown in Tables 4 and 5. Prior to analysis, the samples were blinded.

### ELISA measurement of fetuin-A in serum and CSF

For ELISA studies of fetuin-A concentrations we used the Human Fetuin-A/AHSG DuoSet ELISA (R&D Systems, Minneapolis, USA) as well as 96 well plates and solutions from the DuoSet Ancillary Reagent Kit (R&D Systems, Minneapolis, USA) according to the manufacturers´ recommendations. After dissolving the capture-antibody in 1 ml phosphate buffered saline (PBS) and dilution to 4 µg/ml in PBS, 100 µl of this capture-antibody solution were transferred to every well of the microtiter plate and incubated overnight at 4˚C. The next day the plates were washed twice adding 400 µl PBS-T washing buffer (PBS, 0.05% Tween®20) to every well.

Thereafter the plate was blocked with reagent solution (1% bovine serum albumin (BSA) in PBS, pH 7.2–7.4) adding 300 µl to each well for one hour at room temperature. Thereafter the plates were again washed twice as described above. Prior to transfer to the wells, serum samples were diluted 1:1,000,000 and CSF samples were diluted 1:4,000 in reagent solution. The standards contained in the ELISA kit (see above) were used to prepare a dilution series with

**Table 1. Clinical data of samples for measurement of fetuin-A in CSF and serum with ELISA.**

| | Total (n = 66) | Inflammatory (n = 35) | Non-Inflammatory (n = 27) | Unclear (n = 4) |
|---|---|---|---|---|
| Age at investigation (years)* | 12.49 ± 4,63 (range 1.55–17.88) | 12.15 ± 4.51 (range 2.48–17.88) | 12.80 ± 4.97 (range 1.55–17.65) | 13.44 ± 2.56 (range 10.58–16.57) |
| **Sex distribution** | | | | |
| Female | 42 | 21 | 18 | 3 |
| male | 24 | 14 | 9 | 1 |
| **Serum protein (g/dl)*** | 73.55 ± 5.99 (n = 66) | 74.17 ± 5.89 (n = 35) | 72.89 ± 6.37 (n = 27) | 72.50 ± 2.60 (n = 4) |
| **CSF protein (g/l)*** | 0.36 ± 0.45 (n = 66) | 0.45 ± 0.60 (n = 35) | 0.26 ± 0.10 (n = 27) | 0.23 ± 0.09 (n = 4) |
| **CSF albumin quotient QAlb*** | 5.94 ± 9.49 (n = 53) | 7.87 ± 12.60 (n = 28) | 3.73 ± 2.06 (n = 22) | 4.03 ± 1.64 (n = 3) |
| **C-reactive protein (CRP)** | | | | ( |
| total number | 64 | 35 | 25 | 4 |
| CRP elevated ($\geq$ 5 mg/l) | 9 | 4 | 4 | 1 |
| CRP negative ($<$ 5 mg/l) | 55 | 31 | 21 | 3 |
| **Blood-CSF border (BCB) disorder** | | | | |
| total number | 53 | 28 | 22 | 3 |
| BCB disrupted | 11 | 8 | 2 | 1 |
| BCB normal | 42 | 20 | 20 | 2 |
| **Intrathecal IgG synthesis** | | | | |
| total number | 48 | 26 | 21 | 2 |
| IgG synthesis present | 10 | 9 | 1 | 0 |
| IgG synthesis absent | 39 | 17 | 20 | 2 |

* data shown as mean ± standard deviation.

reagent solution consisting of seven standards with concentrations between 31.25 pg/ml and 2 ng/ml. Then 200 μl of test solution or standard were transferred to every well and incubated for two hours at room temperature to allow binding of fetuin-A to the capture antibody. This

**Table 2. Clinical diagnoses of samples for measurement of fetuin-A in CSF and serum with ELISA.**

| Inflammatory (n = 35) | | Non-inflammatory (n = 27) | | Unclear (n = 4) | |
|---|---|---|---|---|---|
| diagnosis | n | Diagnosis | n | diagnosis | N |
| ADEM | 5 | ataxia | 1 | exclusion demyelination disease | 1 |
| Anti-NMDA receptor encephalitis | 1 | chronic cephalgia | 11 | exclusion neuroborreliosis | 1 |
| clinically isolated syndrome | 1 | chronic shoulder pain | 1 | exclusion meningitis | 1 |
| demyelinating disease | 1 | dyskinesia | 1 | vestibular neuritis | 1 |
| facial palsy | 10 | epilepsy | 3 | | |
| Guillain-Barré syndrome | 2 | exclusion meningitis | 1 | | |
| Herpes simplex virus encephalitis | 1 | paresthesia | 1 | | |
| MS | 4 | paroxysmal movement disorder | 1 | | |
| neuritis nervi optici | 3 | plexus neuropathy | 1 | | |
| neuroborreliosis | 5 | pseudotumor cerebri | 1 | | |
| neuromyelitis optica | 1 | recurrent febrile seizures | 1 | | |
| Varicella zoster virus encephalitis | 1 | somatization disorder | 2 | | |
| | | tremor | 1 | | |
| | | visual loss | 1 | | |

ADEM acute disseminated encephalomyelitis; MS multiple sclerosis

**Table 3. Clinical data of samples for glycosylation studies.**

|  | Total (n = 23) | Inflammatory (n = 13) | control (n = 10) |
|---|---|---|---|
| Age at investigation (years)* | 12.95 ± 4.09 (range 3.97–17.65) | 11.41 ± 4.64 (range 3.97–17.29) | 14.95 ± 1.82 (range 12.13–17.65) |
| Sex distribution |  |  |  |
| female | 16 | 8 | 8 |
| male | 7 | 5 | 2 |

* data shown as mean ± standard deviation.

was followed by three washing cycles (400 μl washing buffer/well). The detection antibody was dissolved in 1 ml reagent solution and diluted to 200 ng/ml in reagent solution. 100 μl of this detection antibody solution were transferred to each well to allow binding of the detection antibody to fetuin-A, leading to a "capture antibody"–"fetuin-A"—"detection antibody" sandwich. After incubation for 2 hours at room temperature additional washing was performed as described above. 100 μl of Streptavidin-HRP solution (1:200 diluted in reagent solution) were transferred to each well and allowed to bind to the detection antibody. After 20 minutes additional washing was performed. This was followed by adding 100 μl substrate solution to each well (1:1 mixture of Color Reagent A ($H_2O_2$) and Color Reagent B (tetramethylbenzidine)). After 20 minutes of incubation while protected from light 50 μl of Stop solution (2 $NH_2SO_4$) were added to each well and the result was quantified using the Fluostar optima Microplate reader (BMG LABTECH GmbH, Ortenberg, Germany). Color change at 450 nm versus color change at 540 nm reference wavelength was measured. The standards were used to generate a standard line which was then used to determine the respective fetuin-A concentrations. Statistical analysis with multiple linear regression was performed using SPSS statistics (IBM Germany GmbH, Ehningen, Germany). Graphical analysis was performed using SPSS and EXCEL (Microsoft).

## Glycosylation and phosphorylation analyses

To study glycosylation of fetuin-A we established a method, based on digestion of CSF and serum samples with glycosidases combined with sodium dodecyl sulfate polyacrylamide gel electrophoresis (SDS-PAGE) and western blot techniques. Glycosidases were PNGase-F and sialiase-Au from an Enzymatic CarboRelease Kit (QA-Bio, KE-DG01, QA-Bio, Inc., Palm

**Table 4. Clinicial diagnoses of samples for glycosylation and phosphorylation studies.**

| Inflammatory for glycosylation studies (n = 13) | | Inflammatory for phosphorylation studies (n = 33) | | Control for glycosylation and phosphorylation studies (n = 10) | |
|---|---|---|---|---|---|
| diagnosis | n | diagnosis | n | diagnosis | n |
| ADEM | 2 | ADEM | 4 | chronic cephalgia | 5 |
| Herpes simplex virus encephalitis | 1 | Anti-NMDA receptor encephalitis | 1 | epilepsy | 1 |
| MS | 4 | clinically isolated syndrome | 4 | exclusion meningitis | 1 |
| neuroborreliosis | 5 | facial palsy | 10 | exclusion neuroborreliosis | 1 |
| SSPE | 1 | Guillain-Barré syndrome | 2 | somatization disorder | 2 |
|  |  | Herpes simplex virus encephalitis | 1 |  |  |
|  |  | MS | 4 |  |  |
|  |  | neuritis nervi optici | 4 |  |  |

ADEM acute disseminated encephalomyelitis; MS multiple sclerosis; SSPE Sclerosing Subacute Panencephalitis

**Table 5. Clinical data of samples for phosphorylation studies.**

|  | Total (n = 43) | Inflammatory (n = 33) | control (n = 10) |
|---|---|---|---|
| **Age at investigation (years)*** | 12.35 ± 4.65 (range 0.50–18.12) | 11.57 ± 4.95 (range 0.50–18.12) | 14.95 ± 1.82 (range 12.13–17.65) |
| **Sex distribution** |  |  |  |
| female | 31 | 23 | 8 |
| male | 12 | 10 | 2 |
| **Serum protein (g/dl)*** | 73.63 ± 5.82 (n = 41) | 71.38 ± 14.12 (n = 31) | 73.50 ± 5.14 (n = 10) |
| **CSF protein (g/l)*** | 1.16 ± 4.94 (n = 43) | 1.43 ± 5.61 (n = 33) | 0.25 ± 0.08 (n = 10) |
| **CSF albumin quotient QAlb*** | 6.48 ± 11.63 (n = 35) | 7.89 ± 13.10 (n = 27) | 3.40 ± 1.08 (n = 8) |
| **C-reactive protein (CRP)** |  |  |  |
| Total number | 41 | 32 | 9 |
| CRP elevated (≥ 5 mg/l) | 6 | 5 | 1 |
| CRP negative (< 5 mg/l) | 35 | 27 | 8 |
| **Blood-CSF border (BCB) disorder** |  |  |  |
| total number | 34 | 26 | 8 |
| BCB disrupted | 7 | 7 | 0 |
| BCB normal | 27 | 19 | 8 |
| **Intrathecal IgG synthesis** |  |  |  |
| Total number | 32 | 24 | 8 |
| IgG synthesis present | 8 | 8 | 0 |
| IgG synthesis absent | 24 | 16 | 8 |

* data shown as mean ± standard deviation.

Desert, USA). PNGaseF (Peptide-N4-(acetyl-ß-glucosaminyl)-asparagine amidase N-Glycosidase F) cleaves N-acetylglucosamine from Asparagine without affecting the structure of the glycan. Sialidase-Au cleaves sialic acids that are bound to a saccharide chain.

Preliminary tests showed that 10% polyacrylamide gels separated well the cleavage products produced by PNGase-F, Sialidase-Au and O-Glycosidase, and that pretreatment of CSF and serum with sialidase-Au was necessary for digestion of fetuin-A with O-glycosidase, β-galactosidase and glucosaminidase. It was also determined that combined cleavage of CSF and serum with sialidase-Au, O-glycosidase, β-galactosidase and glucosaminidase followed by PNGase-F did not improve on consecutive digestion with sialidase-Au and PNGase-F only. Therefore, the latter combination was used throughout this study to separately assess N-glycosylation (PNGase-F digestion) and sialylation (sialidase-Au digestion). A proteinase inhibitor was added to the final reagent solution to stop proteolytic activity that may be present in sera of patients suffering septicemia [59].

To assess its phosphorylation state, fetuin-A was analysed using Phos-Tag™-polyacrylamide gels (Wako Chemicals GmbH, Neuss, Deutschland) with and without pretreatment with alkaline phosphatase (alkaline phosphatase, bovine, 0.151 U/ml, Sigma Aldrich, St. Louis, USA). Unlike conventional SDS-PAGE, Phos-Tag™-PAGE separated phospho-isomers of fetuin-A, which were detected by immunblotting. We also studied PNGase-F digestion of fetuin-A revealing complete cleavage of N-glycosyl carbohydrate sidechains after three hours.

For glycosylation studies serum samples were diluted 1:100 in ultrapure water. CSF samples were not diluted. Both serum and CSF were mixed with reaction buffer (5x), protease inhibitor and denaturation solution. This mixture was denatured and afterwards mixed with Triton-X. It was then distributed to three reaction tubes, containing no glycosidase, 1 μl PNGase-F or 1 μl sialidase-Au. Thereafter, the solutions were allowed to digest for three hours at 37˚C. After the digestion, 25 μl of solution were taken from each reaction tube and mixed with 5 μl 6x SDS

buffer (10 ml 6x SDS buffer consist of: 7 ml 4x stacking gel buffer (0.5M Tris, 0.4% SDS (pH 6.8) + 3 ml glycerin + 0.93 g dithiothreitol (DTT) + 1g SDS + 1.2 mg bromophenol blue sodium salt). This mixture was briefly boiled (5 minutes, 96˚C) to stop digestion.

For phosphorylation studies serum samples were diluted 1:10 in physiological saline, CSF samples were not diluted. 80 μl of diluted serum or 80 μl CSF were mixed with 10 μl NEBuffer-3 (100 mM NaCl, 50 mM Tris-HCl, 10 mM MgCl2, 1 mM DTT, pH 7,9; New England Biolabs, Ipswich, USA) and 10 μl proteinase-inhibitor. This mixture was split into aliquots and alkaline phosphatase was added or not. The solutions were allowed to digest for one hour at 37˚C. To end the digestion, the mixture was heat denatured as above.

Proteins were separated using SDS-PAGE or Phos-Tag™-PAGE, respectively. For the glyco-sylation studies, SDS-PAGE was performed with Mini-PROTEAN® tetra system (Bio-Rad Laboratories, Inc., Hercules, USA) in SDS-buffer according to Laemmli (25mM Tris, 192 mM glycine, 0.1% SDS) (running time 60 minutes, 20mA), using 10% polyacrylamide gels. For phosphorylation analyses, ready-to-use SuperSep™ Phos-tag™- gels (7.5%, including the collection and separation gel parts, Fujifilm Wako Chemicals, Neuss, Germany) containing $Zn^{2+}$-Ions were applied using XCell II or XCell Surelock electrophoresis system (Invitrogen, Thermo Fisher Scientific, Marietta, USA), filled with running buffer (0.25 mol/l Tris, 1.92 mol/l glycine, 1% SDS).

Both gels from SDS-Page and Phos-Tag™-PAGE were then transferred to nitrocellulose membrane. Semi dry blots were performed using an Owl™ Semidry Electric Blotter HEP-1 (Thermo Fisher Scientific, Marietta, USA) at 150 mA constant current per gel for 60 minutes. Protein transfer was verified by Ponceau-S staining.

Nitrocellulose membranes were incubated for 45 minutes at 37˚C in blocking solution comprising PBS-T (PBS, 0.05% Tween-20) and 5% nonfat dried milk powder. Primary anti-body (polyclonal rabbit anti-human fetuin-A, AS5359, made in house [59]) was diluted 1:1000 in blocking solution and incubated for 45 minutes at 37˚C. Secondary antibody (horseradish peroxidase-coupled polyclonal anti-rabbit immunoglobulins, P0217, DAKO, Agilent, Santa Clara, USA) was diluted 1:5000 in blocking solution and incubated for 45 minutes at 37˚C. Following antibody incubations membranes were washed three times with PBS-T for 5 min-utes. Bound antibody was detected by chemiluminescence in substrate solution (0.1M TRIS/HCl, pH 8.5, 1.25mM 3-aminopthalhydrazide, 0.45mM p-coumaric acid, 0.015% hydrogen peroxide) using ImageQuantTM LAS4000 mini (GE Healthcare, Freiburg, Germany). The amount of phosphorylated fetuin-A among total fetuin-A was quantified using Image J soft-ware (Rasband, http://imagej.nih.gov/ij/). Absolute phosphofetuin-A values were calculated from the proportion of phosphorylated fetuin-A among total fetuin-A and the fetuin-A con-centrations from ELISA measurements. Statistical analysis was performed using SPSS statis-tics. Graphical analysis was performed with SPSS and EXCEL.

## Results

### Fetuin-A concentrations in serum and CSF

**Fetuin-A serum concentrations are decreased in probands with inflammatory CNS dis-eases and elevated C-reactive protein serum levels.** Fetuin-A is a negative acute phase pro-tein [60]. To study whether inflammatory processes might be associated with altered serum fetuin-A concentrations we studied samples from 66 probands (Tables 1 and 2). The mean serum fetuin-A concentration was 0.30 mg/ml ± 0.06. A multiple linear regression model was applied to assess relationships between fetuin-A serum concentrations as dependent variable and age, C-reactive protein concentration (cut off: 5 mg/l) and presence of an inflammatory CNS disorder as influencing factors. Analyzing 60 samples we established that increased

serum C-reactive protein level was associated with lower serum fetuin-A ($p < 0.001$). Also, samples of patients with an inflammatory CNS disorder had lower serum fetuin-A ($p = 0.006$). Fig 1A and 1B depict the serum fetuin-A concentrations of different proband groups as box-plots, distinguished by the C-reactive protein level (Fig 1A) and the presence or absence of an inflammatory CNS disorder (Fig 1B), respectively. The statistical results of the linear regression analysis are shown in S1 Table.

**CSF fetuin-A levels are higher in probands with disturbed blood-CSF barrier function.** To study which factors might influence CSF fetuin-A, we measured CSF fetuin-A levels and post-translational modifications. Mean CSF fetuin-A concentration in all 66 samples was 0.644 µg/ml ± 0.55 and thus roughly 500-fold lower than in serum. Data from 45 probands, including 26 with and 19 without inflammatory CNS diseases were included in the analysis. A multiple linear regression model was applied to assess relationships between CSF fetuin-A level and age at investigation, C-reactive protein concentrations (cut off: 5 mg/l), presence of an inflammatory CNS disorder, blood-CSF barrier function and presence of intrathecal IgG synthesis as confounding factors. A disturbed blood-CSF barrier function was defined at a CSF albumin/serum albumin quotient $QAlb > 5 \times 10^{-3}$ (age $\leq$ 15 years) or $QAlb > 6.5 \times 10^{-3}$ (age $>$ 15 years), respectively. CSF fetuin-A concentrations were higher in probands with disturbed blood-CSF barrier function ($p < 0.001$; Fig 1C) and lower in probands with elevated serum C-reactive protein levels ($p = 0.001$; S1 Fig) (S2 Table). The effect of a blood-CSF barrier dysfunction had a stronger influence on the CSF fetuin-A concentration.

**The serum fetuin-A/serum total protein ratio is lower in probands with intrathecal IgG synthesis and in probands with disrupted blood-CSF barrier function.** To assess whether not only absolute fetuin-A concentrations but also the proportion of fetuin-A in total serum protein might be affected by different factors we performed a multiple linear regression including the serum fetuin-A/serum total protein ratio as dependent variable and the following as influencing variables: presence of CNS inflammation, age, C-reactive protein concentrations (cut-off 5 mg/l), presence of intrathecal IgG synthesis and presence of disturbed blood-CSF barrier function. Data from 45 probands were included. In the presence of intrathecal IgG synthesis, the serum fetuin-A/serum protein ratio decreased significantly ($p = 0.006$; Fig 1D). Additionally, we found lower serum fetuin-A/serum protein ratios with a disturbed blood-CSF barrier ($p = 0.005$, Fig 1E) (S3 Table).

**The CSF fetuin-A/CSF total protein ratio increases with age.** Like before we performed a multiple linear regression analysis to assess the relationship between the CSF fetuin-A/CSF total protein ratio as dependent variable and the following as confounding factors: presence of CNS inflammation, age, C-reactive protein concentrations (cut-off 5 mg/l), presence of intrathecal IgG synthesis and presence of disturbed blood-CSF barrier function. Only age was found to correlate positively with the share of fetuin-A among total CSF fetuin A, displayed as scatterplot in Fig 1F ($p = 0.002$) (S4 Table).

**The CSF fetuin-A/serum fetuin-A quotient increases with age and with increase of the CSF albumin/serum albumin quotient and decreases with elevated C-reactive protein concentration.** The quotients CSF IgG/serum IgG (QIgG) and CSF albumin/serum albumin (QAlb) are used to assess intrathecal synthesis of antibodies and blood-CSF barrier function, respectively [48]. Next, we calculated the CSF fetuin-A/serum fetuin-A quotient (QFet) from samples derived from 45 probands and performed a multiple linear regression analysis using QFet as dependent and the following as influencing variables: presence of CNS inflammation, age at investigation, C-reactive protein concentrations (cut-off 5 mg/l), presence of intrathecal IgG synthesis, presence of disturbed blood-CSF barrier function and QAlb. Fig 1G shows a strong positive correlation between QFet and QAlb ($p < 0.001$) as a scatter plot. Both controls and inflammatory groups are shown separately. Data are plotted double logarithmic as is

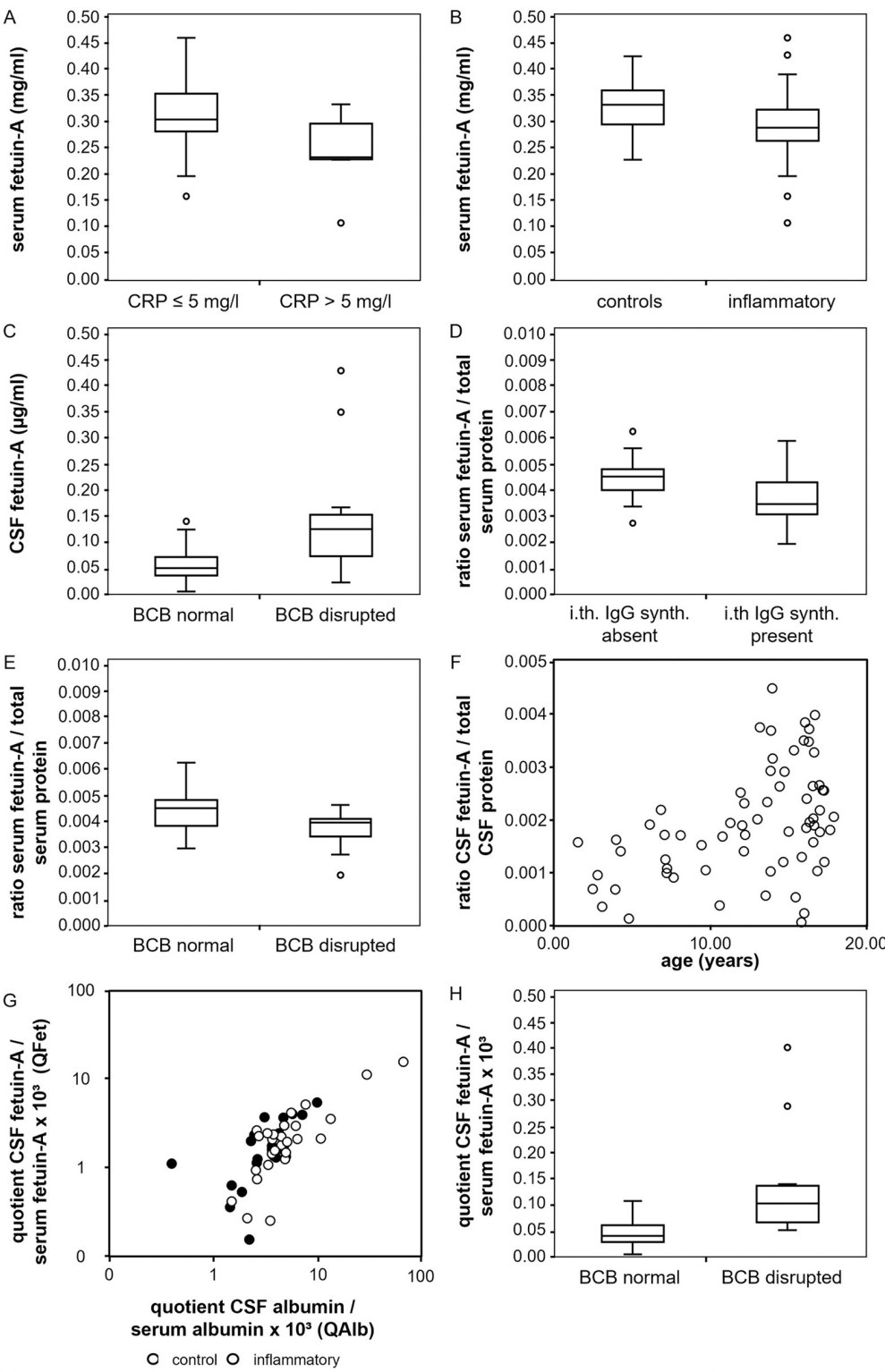

**Fig 1. Concentrations of fetuin-A in serum and CSF.** Fig 1 shows the results of the concentration measurements of fetuin-A in CSF and serum. The correlation of fetuin-A in serum (y-axis, mg/ml) with an elevated C-reactive protein concentration (x-axis, (A)) and the presence of neuroinflammatory disease (x-axis, (B)) is shown. The relation between fetuin-A in CSF (y-axis, μg/ml) and a disruption of the blood-CSF barrier (x-axis) is displayed in (C). (D) and (E) show the connection of the serum fetuin-A/total serum protein ratio (y-axis) with an intrathecal IgG synthesis (x-axis, (D))

and the disruption of the blood-CSF barrier (x-axis, (E)). In (F), the link between the CSF fetuin-A/total CSF protein ratio (y-axis) and age (x-axis, years) is shown as a scatter plot. (G) and (H) show the correlation between the CSF fetuin-A/serum fetuin-A quotient QFet (y-axis, x10$^3$) and the CSF albumin/serum albumin quotient QAlb (x-axis, x10$^3$, (G)) and the presence of a blood-CSF barrier disruption (x-axis, (H)). (G) represents a scatter blot double logarithmically. Values of the control group (black) and inflammatory group (white) are plotted separately. (H) shows a boxplot.

common in Reiber diagrams [48]. Consistent with this finding, an increase of QFet was found in the presence of a blood-CSF barrier disruption (p = 0.001) diagnosed by an elevated QAlb. This is displayed as a boxplot in Fig 1H. In addition, a positive correlation of QFet with age was found (p = 0.007, S1 Fig). An increase with age is also known for QAlb, as described by Reiber [49]. Furthermore, we found a negative correlation between QFet and elevated C-reactive- protein concentration (p = 0.01, S1 Fig).

## Glycosylation patterns of fetuin-A in CSF and serum

Glycosylation was studied in 23 paired CSF/serum samples, including 10 controls and 13 samples derived from patients with inflammatory diseases of the CNS (Tables 3 and 4). SDS-PAGE immunoblotting frequently detected in both serum and corresponding CSF samples a double band indicating the presence of two forms of fetuin-A of different molecular weight. The double band was reduced to a single molecular weight band following treatment of serum samples with sialidase-Au indicating that differential sialylation was responsible for this double band of fetuin-A. The double bands remained unaffected by digestion with PNGase-F. This is shown for four paired CSF/serum samples in Fig 2A (without double bands) and Fig 2B (with double bands).

When comparing the sialylation patterns (double bands) of paired CSF and serum samples from the 23 above mentioned probands, the serum and CSF samples from distinct probands always showed identical sialylation patterns (Fig 2A and 2B) showing that sialylation mostly affected O-linked glycosylation, and was removed by sialidase digestion, but remained unaffected by PNGase-F digestion.

Comparison of sialylation patterns of controls and probands with inflammatory CNS disease revealed that double bands were significantly less frequent among controls (2 out of 10) compared to the inflammatory group (9 out of 13) (Fishers-exact test, two-tailed; p = 0.036). This is depicted as a bar chart in Fig 2C. Our results suggest that desialylated fetuin-A, or asialofetuin-A, is more abundant in inflammatory disorders.

Additionally, we confirmed the presence of two N-glycan carbohydrate chains in fetuin-A [61]. Fig 2D shows that undigested serum fetuin-A (Lane 1) presented as a double band of about 55 and 56 kDa apparent molecular weight in SDS-PAGE/immunoblotting. One minute digestion with PNGase-F resulted in three double bands running at about 55/56, 52/53, and 49/51 kDa representing three different N-glycosylation variants each with or without terminal sialic acid. The 55/56 kDa double band represented undigested 2 N-linked chain fetuin-A, the intermediate 52/53 kDa partially digested single N-glycosylated fetuin-A and the 49/51 kDa completely digested fetuin-A, respectively. After ten minutes of digestion (lane 3) the 55/56 kDa undigested N-glycosylated form of fetuin-A had vanished and after three hours of digestion (lane 4) all fetuin-A had been completely deglycosylated.

## Phosphorylation patterns of fetuin-A in serum and CSF

Phosphorylation patterns of fetuin-A were studied in 10 controls and 33 samples derived from patients with inflammatory CNS diseases (Tables 4 and 5). Phosphorylated fetuin-A (or phosphofetuin-A) was detected in sera of 38 of 41 sera studied. Serum samples with phosphorylated

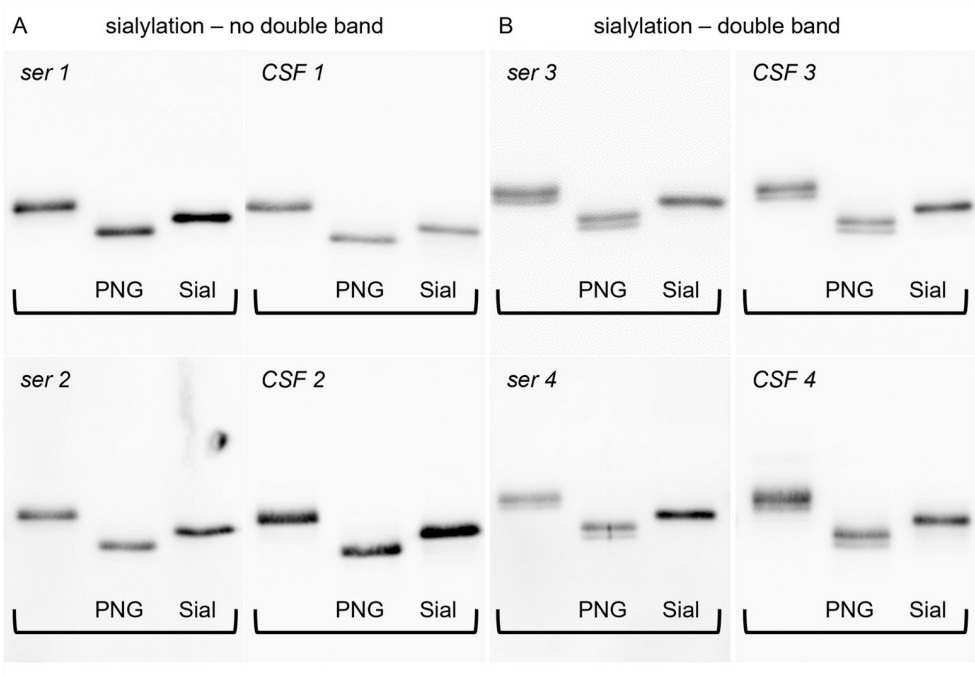

A    sialylation – no double band

B    sialylation – double band

C    sialylation in controls and inflammatory diseases

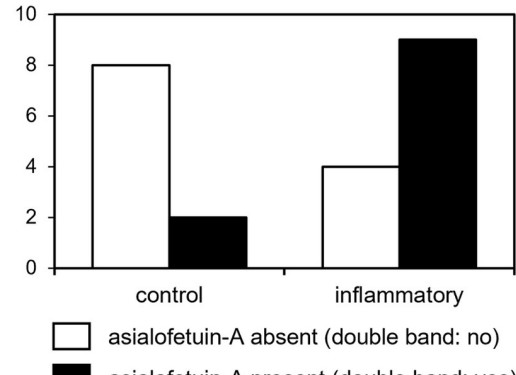

☐ asialofetuin-A absent (double band: no)

■ asialofetuin-A present (double band: yes)

D    N-glycosylation

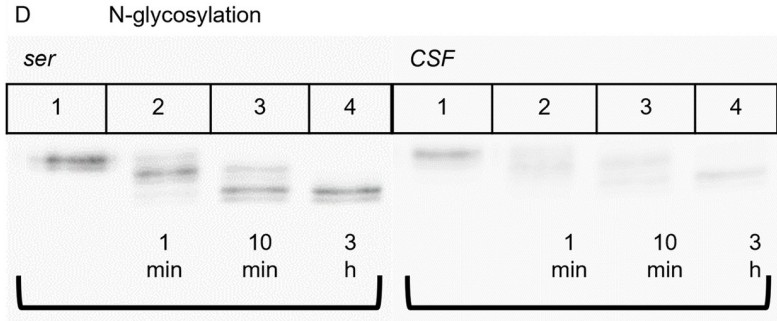

**Fig 2. Glycosylation patterns of fetuin-A in CSF and serum.** Fig 2 illustrates the results of the glycosylation studies of fetuin-A. (A) and (B) show the glycosylation patterns of fetuin-A with (A) and without (B) double bands. On each membrane, lane 1 shows undigested fetuin-A, lane 2 shows fetuin-A after PNGase-F digestion and lane 3 shows fetuin-A after sialidase-Au digestion. (C) compares the occurrence of double bands in the control group and inflammatory group as a bar chart. (D) displays the kinetic of PNGase-F digestion in serum and CSF. Lane 1 shows undigested

fetuin-A, lane 2 shows the pattern after 1 minute of digestion, lane 3 displays the pattern after 10 minutes of digestion and lane 4 shows fetuin-A after 3 hours of PNGase-F digestion.

fetuin-A had a mean percentage of phosphofetuin-A of total serum fetuin-A 8.7% ± 5.8% (range 2% - 28%). The mean absolute concentration of serum phosphofetuin-A was 0.027 mg/ml ± 0.02 mg/ml (range 0.004 mg/ml – 0.076 mg/ml). Compared to serum samples, significantly less CSF samples were found to contain phosphofetuin-A, i.e. only 17 out of 41 samples (Fishers exact test, two-tailed; $p < 0.0001$). In the CSF samples in which phosphofetuin-A was detected, the mean percentage of phosphofetuin-A of total CSF fetuin-A was 7.8% ± 8.2% (range 1%–37%). The mean absolute concentration of CSF phosphofetuin-A was 0.07 μg/ml ± 0.16 μg/ml (range 0.004 μg/ml – 0.716 μg/ml), which is 1000-fold lower than in serum. While glycosylation patterns in CSF and serum from the same proband were identical, this was not evident in phosphorylation. In contrast, there was no correlation of absolute and relative phosphofetuin-A levels between CSF and serum from the same subject. Serum and CSF samples from controls and probands with inflammatory diseases did not differ regarding their absolute and relative serum or CSF phosphofetuin-A levels. However, samples deriving from patients with inflammatory CNS diseases showed higher CSF leukocyte counts (Mann Whitney U test, two tailed; $p < 0.05$) and higher CSF IgG/serum IgG quotients (QIgG) (Mann Whitney U test, two tailed; $p < 0.01$). Absolute phosphofetuin-A levels in CSF were similar in samples with low and high phosphofetuin-A serum levels.

To study the relationship between CSF relative and absolute phosphofetuin-A levels we performed multiple linear regression analyses on the before mentioned CSF samples with relative CSF phosphofetuin-A levels (CSF phosphofetuin-A/CSF total fetuin-A ratio), absolute CSF phosphofetuin-A levels and relative CSF phosphofetuin-A/relative serum phosphofetuin-A quotient as dependent and the following parameters as influencing variables: age at investigation, presence of an inflammatory CNS disorder, CSF leukocyte count, CSF total protein, QAlb, blood-CSF barrier disruption, presence of intrathecal IgG synthesis and serum C-reactive protein level (cut off: 5 mg/l). The relative CSF phosphofetuin-A level was higher with an increase of QAlb as shown as scatter plot in Fig 3A ($p < 0.001$). In the presence of CNS inflammation CSF relative phosphofetuin-A levels were found to be decreased. This is displayed as a boxplot in Fig 3B ($p = 0.016$) (S6 Table). Absolute CSF phosphofetuin-A levels showed an increase with higher QAlb ($p < 0.0001$). This is shown in Fig 3C. Consistent with this finding, absolute CSF phosphofetuin-A levels were increased in the presence of a blood-CSF barrier disruption ($p = 0.003$), which is displayed as a boxplot in Fig 3D. The model summary of the linear regression is shown in S7 Table. Lastly, we investigated the relative CSF phosphofetuin-A/serum phosphofetuin-A quotient. We found an increase of the quotient with an increase of QAlb ($p < 0.0001$). This is shown as a scatter plot in Fig 3E, categorized into control group (black) and neuroinflammatory group (white). In contrast, a decrease of the relative CSF phosphofetuin-A/serum phosphofetuin-A quotient was found in the presence of a neuroinflammatory disorder ($p = 0.014$). This is displayed in Fig 3F. The model summary can be found in S8 Table.

## Discussion

We present a pilot study of serum and CSF fetuin-A including its phosphorylation and glycosylation state in children with and without inflammatory CNS diseases. Fetuin-A is an established negative acute phase protein [60] in various inflammatory diseases including pediatric hemolytic-uremic syndrome, pneumococcal pneumonia [62], chronic inflammatory bowel disease [63], rheumatoid arthritis [64] and septicemia [56]. Corroborating these studies, we

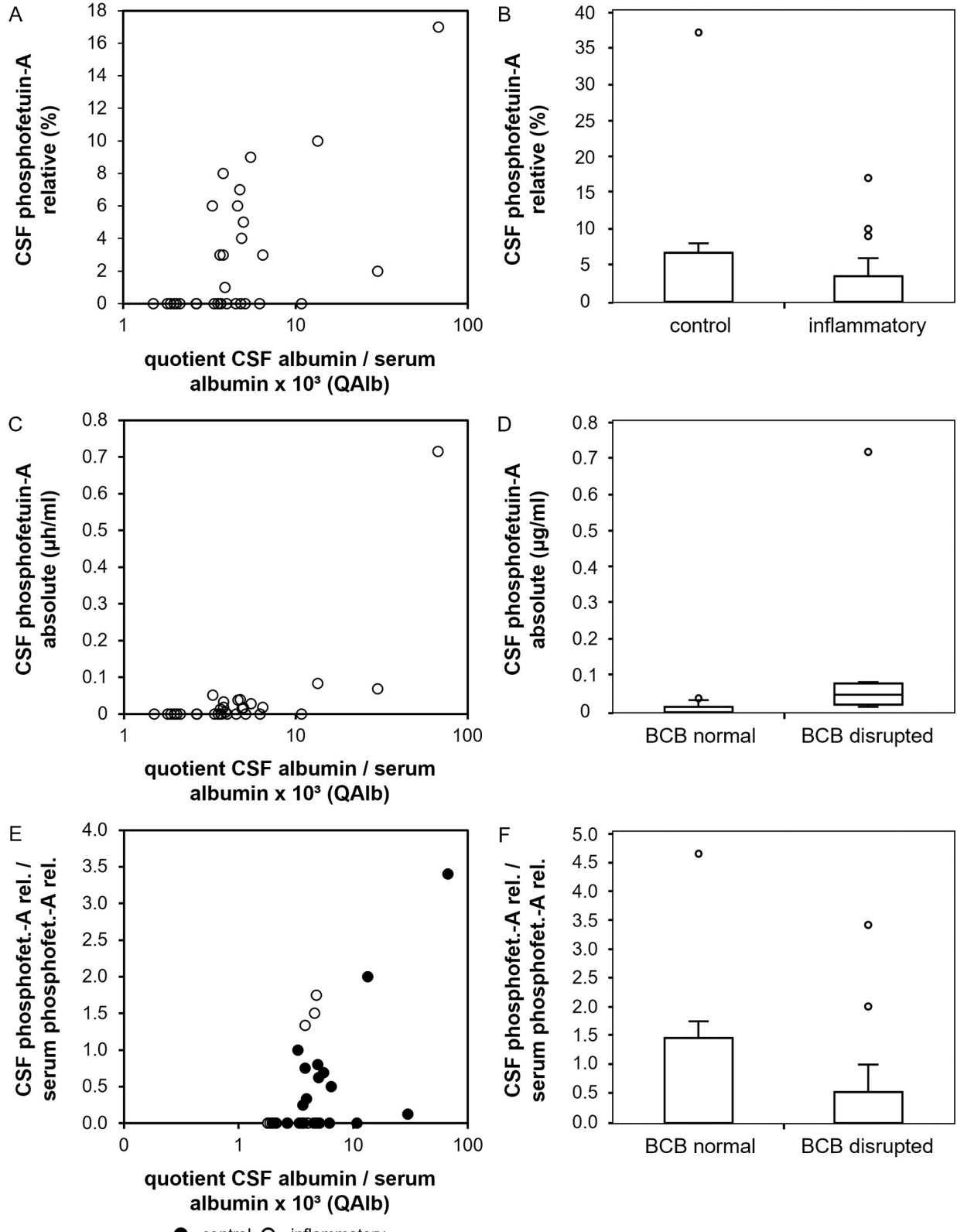

**Fig 3. Phosphorylation patterns of fetuin-A in CSF and serum.** Fig 3 displays the results of the statistical analysis of the phosphorylation studies. (A) and (B) show relative CSF phosphofetuin-A concentrations (y-axis, %) in comparison to the CSF albumin/serum albumin quotient (A, x-axis, x10³) and in the presence of neuroinflammatory diseases (B). (C) and (D) show absolute phosphofetuin-A levels in CSF (y-axis, µg/ml) in comparison to the CSF albumin/serum albumin quotient ((C), x-axis, x10³) and a blood-CSF barrier disruption ((D), x-axis). (E) and (F) display the results of the linear regression with the relative CSF phosphofetuin-A/relative serum phosphofetuin-A quotient. This quotient is shown on the y-axis. (E) shows a scatter plot with the CSF albumin/serum albumin quotient (x-axis, x10³). In (E) the results are shown divided into control group (black) and inflammatory group (white). (F) shows the correlation of the CSF phosphofetuin-A/serum phosphofetuin-A quotient with the presence of a neuroinflammatory disorder (x-axis).

measured reduced serum fetuin-A in probands with increased serum C-reactive protein levels (Fig 1A). The mean serum fetuin-A concentration of 0.30 mg/ml± 0.06 was lower than in our previous studies [2]. This was due to systematic underestimation of fetuin-A levels in serum and CSF by ELISA compared to nephelometry, which was previously used. Our comparison of both methods nevertheless demonstrated strong and linear correlation of nephelometry and ELISA (not shown). Importantly, both methods consistently reported reduced serum fetuin-A in inflammatory disease.

Three mechanisms have been discussed to account for this decrease. First, inhibition of hepatic fetuin-A synthesis by proinflammatory cytokines including TNF-alpha, IL-1beta or IL-6 [3]. Diseases or injuries of the CNS influence hepatic function and thus the synthesis of fetuin-A. This has been reported for spinal cord injuries and other pathologic conditions [65]. Second, sialidase activity in inflammation will produce asialofetuin-A, which is readily cleared by the hepatic asialoglycoprotein receptor (for details see above) [16, 18, 19, 62]. Accordingly, we showed increased asialofetuin-A in neuroinflammation. The third mechanism refers to fetuin-A consumption during inhibition of dystrophic calcification attesting to its general role in tissue chaperoning [66, 67]. Inflammation causes cell damage with release of calcium and phosphate from damaged cells that promotes dystopic calcification while simultaneous energy depletion leads to diminished formation of pyrophosphate [68, 69]. Pyrophosphate, magnesium and fetuin-A are known to collectively prevent extracellular calcification [41]. Fetuin-A, in turn, binds to calcium and phosphate to inhibit dystrophic calcification [35]. Fetuin-A forms calciprotein particles (CPP) which can be cleared [70]. Desialylation may serve as a regulatory signal for the degradation of fetuin-A and its clearance after binding calcium and phosphate.

We report that CSF fetuin-A concentrations were about 500 times lower than serum fetuin-A concentrations with mean levels of about 0.3 mg/ml in serum and 0.6 µg/ml in CSF, respectively. These CSF- and serum levels are lower than in fetal CSF or serum but in a similar range as previously reported for adults [2, 54, 55, 71]. Age-dependent reference values are today not established. Further research is required with a larger sample size.

Like in systemic inflammation, serum fetuin-A was also lower in patients with inflammatory CNS disease (Fig 1B). Moreover, the proportion of serum fetuin-A among total serum protein proved lower in probands with disrupted blood-CSF barrier and in probands with intrathecal IgG synthesis (Fig 1D and 1E). This indicates an independent and yet unexplained effect of CNS inflammation on systemic fetuin-A levels. CSF fetuin-A levels were reported in different human diseases. This includes reduced CSF levels indicative of progression of clinically isolated syndrome to definitive MS [72], increased CSF levels in patients with low grade gliomas [73] and in patients with secondary progressive MS [74] as well as decreased CSF fetuin-A in patients with Alzheimer´s disease [58]. Only few studies, however, addressed the relationship between CSF fetuin-A concentrations, blood-CSF barrier function and CNS inflammation, for example focusing on CSF and serum samples deriving from adults with multiple sclerosis. Harris et al. found higher CSF fetuin-A concentrations and higher CSF fetuin-A/plasma fetuin-A quotients in patients with active compared to inactive multiple

sclerosis whereas both groups did not differ regarding their serum fetuin-A levels. As the CSF albumin/serum albumin quotients were similar in patients with active and inactive MS the authors concluded that higher CSF fetuin-A levels in active MS resulted from intrathecal fetuin-A synthesis rather than from entry of fetuin-A into the CNS via the blood-CSF barrier [54]. In a further study using an EAE mouse model Harris et al. found reduced disease severity in fetuin-A-deficient mice suggesting that fetuin-A might actively contribute to the immune response in cerebral autoimmune disorders. They further reported on fetuin-A expression of mouse microglial cells after LPS stimulation which points to the possibility of intrathecal fetuin-A synthesis [55]. By analyzing paired proteomes from plasma and CSF in older patients, Dayon and colleagues reported a strong correlation between blood-CSF barrier disturbance and increased CSF fetuin-A levels [75].

We found decreased CSF fetuin-A in probands with increased serum CRP, but increased CSF fetuin-A if blood-CSF barrier function was impaired (Fig 1C). This argues for passive entry of fetuin-A into the CNS during blood-CSF barrier disruption and a simultaneous down-regulation related to systemic inflammation. The ratio of CSF fetuin-A to total CSF protein, in turn, was chiefly and positively related to the probands´ age but independent from blood-CSF barrier function and from intrathecal immunoglobulin synthesis (Fig 1G). This observation again supports passive entry of fetuin-A into the CNS together with further serum proteins. The high impact of age on CSF fetuin-A levels reflects the fact that CSF levels of serum proteins decrease between birth and 6 months of age, followed by a slight age-related increase (trans-thyretin, albumin, alpha2-proteins and gamma globulins) [76]. This delayed increase in dis-tinct CSF proteins may be related to the physiological age-related increase of blood-CSF barrier permeability that can be assumed from the known increase of the CSF albumin/serum albumin quotient with age [48, 49].

Like total CSF fetuin-A levels, the CSF fetuin-A/serum fetuin-A quotient increased with age. This quotient was also increased during blood-CSF barrier dysfunction or higher CSF albumin/serum albumin quotients, again supporting an increased and passive influx of fetuin-A into the CNS during CNS inflammation. In contrast to the above mentioned physiological age-related increase of the CSF albumin/serum albumin quotient [48, 49], a comparable age-related increase of this quotient was not present in the here studied pediatric samples. There-fore, the age-related increase of the CSF fetuin-A/serum fetuin-A quotient cannot be attributed to an age-related increase of blood-CSFbarrier permeability in the pediatric age group. Alter-native explanations might be diminished fetuin-A degradation in the brain compartment or increased fetuin-A degradation in the systemic compartment with increase of age.

The apparent molecular weight of fetuin-A (about 60 kD) is slightly lower than the molecu-lar weight of albumin (about 66 kD). Therefore, the CSF fetuin-A/serum fetuin-A quotient should change in a similar fashion as the CSF albumin/serum albumin quotient (Fig 1H). On the one hand the CSF fetuin-A/serum fetuin-A quotients measured in this study showed an increase with an increase of the CSF albumin/serum albumin quotient as typical for serum-derived proteins entering the CSF. On the other hand, this increase was much slower than anticipated. This raises the question whether fetuin-A may be retained at the blood-CSF bar-rier or actively consumed in the intrathecal compartment [49].

We confirmed two patterns of glycosylation of fetuin-A comprising two-N-bound glycans and sialylation. As desialylation removed the double fetuin-A bands on western blots this observation was studied in more detail. Double bands were either present or absent from paired serum/CSF samples of distinct probands (Fig 2A and 2B). This indicates that desialyla-tion does not restrict fetuin-A from entering the CNS via the blood-CSF barrier. It also makes an intrathecal synthesis of fetuin-A unlikely as this might lead to different sialylation patterns of fetuin-A in serum and CSF. As already mentioned above, asialofetuin-A is rapidly cleared

from circulation after binding to the hepatic asialoglycoprotein receptor [18, 19]. Among the here studied samples, double bands, i.e. asialofetuin-A, were predominantly detected in the context of inflammatory CNS disorders. This confirms previous studies reporting increased asialofetuin-A in patients with inflammatory diseases. Possibly, in inflammatory conditions fetuin-A experiences a qualitative and quantitative downregulation, mediated by enhanced degradation, diminished functional capabilities, diminished hepatic synthesis and consumption related to calcium release during cell damage.

In contrast to glycosylation, fetuin-A phosphorylation patterns of CSF and serum specimen from distinct probands were not equal, indicating a more complex regulation of phosphorylation in different body compartments. Paired data on CSF and serum phosphofetuin-A were available from 41 probands. None of the three samples negative for phosphofetuin-A in serum was positive for phosphofetuin-A in CSF whereas 19 of the 38 samples positive for phosphofetuin-A in serum were negative for phosphofetuin-A in CSF. On the one hand this might indicate that the metabolically active phosphorylated fetuin-A has restricted access to the CNS, on the other hand this might point to consumption of fetuin-A within the nervous system. However, like global fetuin-A levels in CSF, the strongest predictor for high relative and absolute phosphofetuin-A levels in CSF was the CSF albumin/serum albumin quotient QAlb, which again argues in favor of passive entry of phosphofetuin-A into the CSF. Absolute phosphofetuin-A levels in CSF were higher in probands with blood-CSF barrier disruption, which indicates that phosphorylated fetuin-A also gets access to the CNS in a passive fashion rather than being synthesized in the CNS itself.

Limitations of our pilot study regard the small number of samples, which is insufficient to firmly establish fetuin-A as a marker of CNS inflammation, let alone subtypes thereof. We are aware that the analysis by (Phos-tag) SDS-PAGE and Western-Blot of PTM-fetuin-A also is semi-quantitative at best and could be greatly improved by protein-mass spectrometry of serum and CSF samples. This would however require internal standards, preferably labelled fetuin-A standards, which are not currently freely available. Future studies will need to address these issues.

## Summary

In this study we investigated serum and CSF fetuin-A including its glycosylation and phosphorylation state in samples of pediatric patients with and without neuroinflammatory disorders. We report lower serum fetuin-A concentrations in inflammation consistent with its nature as a negative acute phase protein. Our study showed evidence of a passive influx of fetuin-A into the CNS, especially in CNS inflammation. Asialofetuin-A was found more frequent in probands with CNS inflammation, suggesting a possible regulatory function of desialylation for fetuin-A clearance after formation of CPP. We found no differences between serum and CSF samples of each patient. In contrast, phosphofetuin-A was more abundant in serum samples than in CSF, indicating a regulation of influx of fetuin-A over the blood-CSF barrier by phosphorylation. Our findings may help to better understand the function of fetuin-A in CNS inflammation and its potential as a biomarker of neuroinflammatory diseases.

## Supporting information

**S1 Fig. Concentrations of fetuin-A in CSF and CSF fetuin-A/serum fetuin-A quotient.** S1 Fig shows supplementary results of the concentrations measurements of fetuin-A in CSF and serum. The correlation of CSF fetuin-A (y-axis, μg/ml) with an elevated C-reactive protein concentration (x-axis) is displayed as a boxplot in (A). The connection between the CSF fetuin-A/serum fetuin-A quotient (y-axis, $x10^3$) and age (x-axis, years) is displayed as scatter

plot in (B). (C) shows the correlation of the CSF fetuin-A/serum fetuin-A quotient (y-axis, $x10^3$) with an elevated C-reactive protein concentration (x-axis) as a boxplot.
(TIF)

**S1 Table. Multiple linear regression.** Predictors for fetuin-A concentration (mg/ml) in serum.
(PDF)

**S2 Table. Multiple linear regression.** Predictors for fetuin-A concentration (μg/ml) in cerebrospinal fluid.
(PDF)

**S3 Table. Multiple linear regression.** Predictors for serum fetuin-A/serum total protein ratio.
(PDF)

**S4 Table. Multiple linear regression.** Predictors for CSF fetuin-A/CSF total protein ratio.
(PDF)

**S5 Table. Multiple linear regression.** Predictors for the CSF fetuin-A/serum fetuin-A quotient.
(PDF)

**S6 Table. Multiple linear regression.** Predictors for relative CSF phosphofetuin-A concentrations.
(PDF)

**S7 Table. Multiple linear regression.** Predictors for absolute CSF phosphofetuin-A concentrations.
(PDF)

**S8 Table. Multiple linear regression.** Predictors for relative CSF phosphofetuin-A/relative serum phosphofetuin-A quotient.
(PDF)

**S1 Raw images. Raw images for Fig 2.**
(PDF)

**S1 Data. Raw data for the concentration, glycosylation and phosphorylation analysis.**
(XLSX)

## Author Contributions

**Conceptualization:** Frederik Ricken, Martin Häusler, Willi Jahnen-Dechent.

**Data curation:** Frederik Ricken, Ahu Damla Can, Steffen Gräber, Martin Häusler.

**Formal analysis:** Frederik Ricken, Ahu Damla Can, Martin Häusler.

**Funding acquisition:** Martin Häusler, Willi Jahnen-Dechent.

**Investigation:** Frederik Ricken, Ahu Damla Can, Steffen Gräber.

**Project administration:** Willi Jahnen-Dechent.

**Resources:** Martin Häusler, Willi Jahnen-Dechent.

**Supervision:** Martin Häusler, Willi Jahnen-Dechent.

**Validation:** Martin Häusler, Willi Jahnen-Dechent.

**Visualization:** Frederik Ricken, Ahu Damla Can.

**Writing – original draft:** Frederik Ricken, Martin Häusler.

**Writing – review & editing:** Frederik Ricken, Ahu Damla Can, Martin Häusler, Willi Jahnen-Dechent.

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
