## [Decision Letter · Decision Letter 0]

11 Jul 2022

PONE-D-22-12868Post-translational modifications glycosylation and phosphorylation of the major hepatic plasma protein fetuin-A are associated with severity of CNS inflammation in childrenPLOS ONE

Dear Dr. Jahnen-Dechent,

Thank you for submitting your manuscript to PLOS ONE. After careful consideration, we feel that it has merit but does not fully meet PLOS ONE’s publication criteria as it currently stands. Therefore, we invite you to submit a revised version of the manuscript that addresses the points raised during the review process. As you can see, both reviewers appreciated your work, but also came up with numerous suggestions how to further improve it.

We look forward to receiving your revised manuscript.

Kind regards,

Pavel Strnad

Academic Editor

PLOS ONE

Journal Requirements:

  "This research project is supported by the START-Program of the Faculty of Medicine of the RWTH Aachen University (grant number 129/14) (MH) and the German Research Foundation (DFG, TRR 219, Project ID 322900939) (WJD)."

Reviewers' comments:

Reviewer's Responses to Questions

**Comments to the Author**

1. Is the manuscript technically sound, and do the data support the conclusions?

Reviewer #1: Yes

Reviewer #2: Partly

2. Has the statistical analysis been performed appropriately and rigorously? 

Reviewer #1: Yes

Reviewer #2: Yes

3. Have the authors made all data underlying the findings in their manuscript fully available?

Reviewer #1: Yes

Reviewer #2: No

4. Is the manuscript presented in an intelligible fashion and written in standard English?

Reviewer #1: Yes

Reviewer #2: No

5. Review Comments to the Author

Reviewer #1: The authors of this manuscript have done an exceptional job describing the complexities of CNS inflammation and demonstrated the usefulness of Fetuin-A as a potential biomarker of inflammation. Readers will be well equipped with the experimental details given in this manuscript to study Fetuin-A and its glycosylation and phosphorylation.

I would like to recommend the following additional information that could be added to the manuscript:

1. Age and sex distribution of the study cohorts including controls be provided if not already

2. Provide an explanation of dietary influence on Fetuin-A and how your study controlled that influence

3. Provide CSF sample collection details. Explain how the effects of rostrocaudal gradient on CSF Fetuin-A was controlled in this study.

4. Provide details on the inflammatory conditions and controls. A table with patient demographics and clinical conditions would be helpful.

5. Explain if Fetuin-A can discriminate between different CNS inflammatory/autoimmune conditions as well as Systemic inflammation Vs CNS inflammation.

6. Age specific reference intervals for Fetuin-A should be the next step once a larger data set is available

Reviewer #2: The Authors studied CSF and serum concentrations of total as well as post-translationally modified fetuin-A in a group of 66 children (including controls and various CNS inflammatory diseases). The aim of the study is excellently described on p. 8, rows 140-144. They came to the conclusion that 1) fetuin-A in the CSF is blood-derived rather than being synthesised intrathecally (probably the most important result of the study); 2) asialofetuin-A was found more frequently in patients with CNS inflammation; 3) phosphofetuin-A was more abundant in serum samples than in CSF, possibly indicating a regulation of influx of fetuin-A over the blood-CSF barrier by phosphorylation.

In my opinion, the study has been executed very carefully and is of significant theoretical importance. The main strength of the study lies in the quantitative assessment (within the methodological limits of the assays applied) of both total and phosphorylated protein forms, a concept that should perhaps be applied to several other proteins (and maybe other post-translational modifications) when studying the origin of CSF proteins (blood-derived vs. intrathecally synthesised). Qualitative analysis of glycosylation patterns in CSF and serum is also very interesting. Many interesting aspects of fetuin-A physiology and possible relevance for CNS inflammation are carefully reviewed by the Authors that apparently are experts on fetuin-A biology. However, I have not been convinced by their arguments that CSF fetuin-A measurement can be of any significant clinical value in routine practice (there is no intrathecal fetuin-A synthesis; decreased concentrations and relatively increased asialofetuin-A were found by the Authors also in serum, so why to investigate CSF for this protein?) In this context, I would suggest to reconsider an appropriate manuscript title (also see below under Major criticisms, point 1).

In my opinion, the Manuscript deserves Major revision before acceptance. Major criticisms are:

1) clinical data of the subjects are not provided, except for partial information on page 17 (lines 339-343) concerning samples for glycosylation analysis. This should be definitely corrected. Definition of control group and diagnoses of patients in the inflammatory group should be provided in the Methods section under the Heading Probands. Since number of samples analyzed by various assays (total fetuin-A, glycosylation, phosphorylation) differs, number of patients in both groups (controls and CNS inflammation) should be provided separately for each assay. Looking at diagnoses of a subgroup mentioned at p.17, this is a mix of various CNS inflammatory diseases with substantially different CSF profiles and different immunopathogenesis, most of them usually associated with normal serum CRP levels. However, this is acceptable in a pilot study.

How was the severity of CNS inflammation assessed? This is important if the Authors wish to retain in the Article title that post-translational modifications of fetuin-A are associated with SEVERITY (rather than presence) of CNS inflammation in children.

Basic information about methods used for CSF analysis, especially total protein, albumin and IgG should be provided as well (method - e.g. turbidimetric or dye-binding for total protein, reagent kit, and instrument).

2) Throughout the text, the Authors should consider using the term "Blood-CSF barrier" instead of "Blood-brain barrier", especially in the context of CSF/Serum protein quotients.

3) The term "ratio" could possibly be replaced by "quotient" as the latter is used by most authorities in CSF analysis, including prof. H. Reiber and prof. E.Thompson. In Thompson´s book Proteins of the cerebrospinal fluid (Elsevier 2005, ISBN 0-12-369369-1), a review of appropriate terms is provided on pp. 10-11: Ratio is defined as the result of dividing the amount of one protein (e.g. IgG) in CSF by the amount of another (e.g. albumin) also in the CSF, whereas Quotient is the result of dividing the amount of a given protein (e.g. albumin) in the CSF by the amount of the same protein again (e.g. albumin) in the serum.

4) Throughout the Text and Figures, axis label for CSF/Serum quotients should contain x10^3 and NOT 10^-3 E.g., for CSF fetuin-A 0,6 mg/L and Serum fetuin-A 0,3 g/L = 300 mg/L, the CSF/S quotient is 0,002 - this should be multiplied by a factor of 1000 if we want to get convenient numbers (in this case, 2) on the x-axis.

5) Figure 1G (CSF/S Q of total Fetuin-A on the y axis versus Q Alb on the x axis) seems to me to be perhaps the most important. Please try to indicate 2 patient subgroups (controls vs. CNS inflammation) in this Figure by using different marks or colours for these 2 patient groups. The same might be of value for Fig. 3E. (Concerning Fig. 3E, I am not sure whether absolute values of phosphofetuin-A CSF/Serum concentration quotients could be more appropriate than relative values on the y-axis.)

Although not obligatory, I would recommend at least briefly discuss in the Discussion section: 1) why CSF/serum fetuin A quotients are somewhat lower than anticipated based on the molecular weight of this protein (compared with albumin with M.w. of 67 kDa, fetuin-A should have CSF/serum quotients close to Q albumin or somewhat higher). Since the CSF/S quotients depend rather on hydrodynamic radius than on M.w., have the Authors any information about the hydrodynamic radius of fetuin-A compared to albumin? (partly discussed on p. 25 rows 516-523, but many Q fetuin-A values are much lower than corresponding Q Alb values also in patients with normal Q Alb), 2) the Authors tried to perform some kind of quantitation of the phosphorylated fetuin-A; what about the precision of these data (%CV in repeated experiments or from the literature etc.), and do the Authors think such quantitative attempts are possible and worthwhile for their glycosylation studies? 3) The Authors used enzyme digestion to study post-translational modifications of fetuin-A. It would be interesting if they wish at least to mention the use of mass spectrometry for such a purpose and why they did not choose this analytical tool in their study (not suitable? not available? other reason?)

Minor Criticisms:

Abstract, p.2, row 23: MS - abbreviation not explained (multiple sclerosis?), I would recommend to drop it from the Abstract since the role of fetuin-A as a biomarker for MS is far from being firmly established.

p.4 row 52 and p.22 row 449: Greek symbols or letters should be used consistenly for TNF-alpha and IL-1beta throughout the text

p.10 row 188 - "multiple" instead of "multipe"

p.10 - subscript should be used appropriately in the formulas of chemical compounds (e.g. for hydrogen peroxide)

p. 11 row 192: "sodium dodecyl sulfate" instead of "sodiumdodecylsulfat"; row 208: "polyacrylamide" instead of "polyacrylamid"

p.13 row 237: "2+" should be in superscript

p.13 row. 246 please provide brief description of the in-house AS5359 anti-human fetuin-A antibody, or provide an appropriate reference in the Reference list

p. 16 rows 307-308: it is not clear what the Authors wished to say, and it seems to be related to Fig. 1F rather than Fig. 1G; please try to formulate in a clearer way

p. 16 row 317: Fig. 1G rather than 1H; row 321: Fig. 1H rather than 1G

p. 17 rows 339-343: this should be stated in the Methods section (see above)

p.19 row 392: incorrect unit for the range of CSF phospho fetuin-A concentrations (mg/ml instead of micrograms/ml)

p. 22 row 444: please use "CSF" instead of "liquor"

p. 26 row 533: "in the context" rather than "in the contexts"

Tables S6 and S7 have identical Table title; I would recommend either to use different Table titles or to merge in one Table

6. PLOS authors have the option to publish the peer review history of their article (what does this mean?). If published, this will include your full peer review and any attached files.

Reviewer #1: No

Reviewer #2: No

---

## [Author Response · Author response to Decision Letter 0]

5 Sep 2022

The following text is also contained in the file "response to reviewers" where it is formatted for better reading.

Reviewers' comments:

Reviewer's Responses to Questions

Comments to the Author

1. Is the manuscript technically sound, and do the data support the conclusions?

Reviewer #1: Yes

Reviewer #2: Partly

2. Has the statistical analysis been performed appropriately and rigorously?

Reviewer #1: Yes

Reviewer #2: Yes

3. Have the authors made all data underlying the findings in their manuscript fully available?

The PLOS Data policy requires authors to make all data underlying the findings described in their manuscript fully available without restriction, with rare exception (please refer to the Data Availability Statement in the manuscript PDF file). The data should be provided as part of the manuscript or its supporting information or deposited to a public repository. For example, in addition to summary statistics, the data points behind means, medians and variance measures should be available. If there are restrictions on publicly sharing data—e.g. participant privacy or use of data from a third party—those must be specified.

Reviewer #1: Yes

Reviewer #2: No

We now fully share all data as an anonymized data set contained in the Supporting Information file “S1_raw data”. All uncropped and unadjusted blot images are provided in the Supporting Information file “S1_raw_images”.

4. Is the manuscript presented in an intelligible fashion and written in standard English?

Reviewer #1: Yes

Reviewer #2: No

We thank reviewer for pointing out several typographical or grammatical errors, which we have now corrected.

5. Review Comments to the Author

Reviewer #1: The authors of this manuscript have done an exceptional job describing the complexities of CNS inflammation and demonstrated the usefulness of Fetuin-A as a potential biomarker of inflammation. Readers will be well equipped with the experimental details given in this manuscript to study Fetuin-A and its glycosylation and phosphorylation.

Thank you very much for this overall very positive feedback.

I would like to recommend the following additional information that could be added to the manuscript:

1. Age and sex distribution of the study cohorts including controls be provided if not already

2. Provide an explanation of dietary influence on Fetuin-A and how your study controlled that influence

3. Provide CSF sample collection details. Explain how the effects of rostrocaudal gradient on CSF Fetuin-A was controlled in this study.

4. Provide details on the inflammatory conditions and controls. A table with patient demographics and clinical conditions would be helpful.

5. Explain if Fetuin-A can discriminate between different CNS inflammatory/autoimmune conditions as well as Systemic inflammation Vs CNS inflammation.

6. Age specific reference intervals for Fetuin-A should be the next step once a larger data set is available

I would like to recommend the following additional information that could be added to the manuscript:

1. Age and sex distribution of the study cohorts including controls be provided if not already

The requested information is now available for three cohorts concentration, glycosylation, and phosphorylation.

2. Provide an explanation of dietary influence on Fetuin-A and how your study controlled that influence

Unfortunately, dietary influence was not study or controlled.

3. Provide CSF sample collection details. Explain how the effects of rostrocaudal gradient on CSF Fetuin-A was controlled in this study. 

We agree that concentrations of different CSF components may differ depending on whether CSF was obtained directly from the ventricles or from CSF obtained during lumbar punctures. All samples of this investigation had been obtained by lumbar punctures, which is the standard method to obtain cerebrospinal fluid in children for diagnostic purposes. Therefore, the rostrocaudal gradient of CSF cannot account for differences observed between groups.

4. Provide details on the inflammatory conditions and controls. A table with patient demographics and clinical conditions would be helpful. 

Information is now provided as tables of anonymous data in the Materials/Probands section. Serum chemistry, clinical data, and diagnosis available to us are all included.

5. Explain if Fetuin-A can discriminate between different CNS inflammatory/autoimmune conditions as well as Systemic inflammation Vs CNS inflammation.

This is indeed an interesting point. Depending on whether there is systemic inflammation or CNS inflammation, fetuin-A expression may indeed change as we have recently observed in kidney fetuin expression under hypoxia. Healthy kidney does not produce fetuin-A mRNA, but hypoxic kidney does (ref. 67). In pathological conditions, expression of fetuin-A mRNA may also be induced in the choroid plexus, an established extrahepatic fetuin-A mRNA expressing tissue. As much as we would like to clarify this issue, it is beyond the scope of this pilot study due to limited quality and quantity of available material. We mention this as a limitation of our study in the discussion.

6. Age specific reference intervals for Fetuin-A should be the next step once a larger data set is available. 

Again, we fully agree. Like the differentiation of systemic vs. CNS inflammation this requires a larger number of samples.

Reviewer #2: The Authors studied CSF and serum concentrations of total as well as post-translationally modified fetuin-A in a group of 66 children (including controls and various CNS inflammatory diseases). The aim of the study is excellently described on p. 8, rows 140-144. They came to the conclusion that 1) fetuin-A in the CSF is blood-derived rather than being synthesised intrathecally (probably the most important result of the study); 2) asialofetuin-A was found more frequently in patients with CNS inflammation; 3) phosphofetuin-A was more abundant in serum samples than in CSF, possibly indicating a regulation of influx of fetuin-A over the blood-CSF barrier by phosphorylation.

In my opinion, the study has been executed very carefully and is of significant theoretical importance. The main strength of the study lies in the quantitative assessment (within the methodological limits of the assays applied) of both total and phosphorylated protein forms, a concept that should perhaps be applied to several other proteins (and maybe other post-translational modifications) when studying the origin of CSF proteins (blood-derived vs. intrathecally synthesised). Qualitative analysis of glycosylation patterns in CSF and serum is also very interesting. Many interesting aspects of fetuin-A physiology and possible relevance for CNS inflammation are carefully reviewed by the Authors that apparently are experts on fetuin-A biology. However, I have not been convinced by their arguments that CSF fetuin-A measurement can be of any significant clinical value in routine practice (there is no intrathecal fetuin-A synthesis; decreased concentrations and relatively increased asialofetuin-A were found by the Authors also in serum, so why to investigate CSF for this protein?) In this context, I would suggest to reconsider an appropriate manuscript title (also see below under Major criticisms, point 1).

In my opinion, the Manuscript deserves Major revision before acceptance. Major criticisms are:

We would like to thank reviewer 2 for the interest taken in our work, the meticulous review, and the many good suggestions how to improve the manuscript. We gladly accepted all changes suggested by the reviewer.

1) clinical data of the subjects are not provided, except for partial information on page 17 (lines 339-343) concerning samples for glycosylation analysis. This should be definitely corrected. Definition of control group and diagnoses of patients in the inflammatory group should be provided in the Methods section under the Heading Probands. Since number of samples analyzed by various assays (total fetuin-A, glycosylation, phosphorylation) differs, number of patients in both groups (controls and CNS inflammation) should be provided separately for each assay. Looking at diagnoses of a subgroup mentioned at p.17, this is a mix of various CNS inflammatory diseases with substantially different CSF profiles and different immunopathogenesis, most of them usually associated with normal serum CRP levels. However, this is acceptable in a pilot study.

We apologize for being so terse in the first version of our manuscript. This information is now provided as tables of anonymous data in the Materials/Probands section. Serum chemistry, clinical data and diagnosis available to us

How was the severity of CNS inflammation assessed? This is important if the Authors wish to retain in the Article title that post-translational modifications of fetuin-A are associated with SEVERITY (rather than presence) of CNS inflammation in children.

We agree that we should tone down on claims about fetuin-a and severity of CNS inflammation. We deleted this from the title.

Basic information about methods used for CSF analysis, especially total protein, albumin and IgG should be provided as well (method - e.g. turbidimetric or dye-binding for total protein, reagent kit, and instrument).

All routine clinical chemistry assays were performed by the clinical chemistry laboratory of RWTH Aachen Clinics. Information obtained from the lab is now included in Methods. Fetuin-A analyses were performed in our own laboratory. 

2) Throughout the text, the Authors should consider using the term "Blood-CSF barrier" instead of "Blood-brain barrier", especially in the context of CSF/Serum protein quotients.

We thank the reviewer for this valuable suggestion. Done throughout!

3) The term "ratio" could possibly be replaced by "quotient" as the latter is used by most authorities in CSF analysis, including prof. H. Reiber and prof. E.Thompson. In Thompson´s book Proteins of the cerebrospinal fluid (Elsevier 2005, ISBN 0-12-369369-1), a review of appropriate terms is provided on pp. 10-11: Ratio is defined as the result of dividing the amount of one protein (e.g. IgG) in CSF by the amount of another (e.g. albumin) also in the CSF, whereas Quotient is the result of dividing the amount of a given protein (e.g. albumin) in the CSF by the amount of the same protein again (e.g. albumin) in the serum.

We changed the wording quotient and ratio as throughout as suggested by reviewer.

4) Throughout the Text and Figures, axis label for CSF/Serum quotients should contain x10^3 and NOT 10^-3 E.g., for CSF fetuin-A 0,6 mg/L and Serum fetuin-A 0,3 g/L = 300 mg/L, the CSF/S quotient is 0,002 - this should be multiplied by a factor of 1000 if we want to get convenient numbers (in this case, 2) on the x-axis.

Done as suggested.

5) Figure 1G (CSF/S Q of total Fetuin-A on the y axis versus Q Alb on the x axis) seems to me to be perhaps the most important. Please try to indicate 2 patient subgroups (controls vs. CNS inflammation) in this Figure by using different marks or colours for these 2 patient groups. The same might be of value for Fig. 3E. (Concerning Fig. 3E, I am not sure whether absolute values of phosphofetuin-A CSF/Serum concentration quotients could be more appropriate than relative values on the y-axis.)

Done as suggested. Absolute values of phosphofetuin-A cannot be obtained with phostag-PAGE and immunoblotting. This will have to await protein-MS with internal (labeled) standards. We mention this in the discussion.

Although not obligatory, I would recommend at least briefly discuss in the Discussion section: 

1) why CSF/serum fetuin A quotients are somewhat lower than anticipated based on the molecular weight of this protein (compared with albumin with M.w. of 67 kDa, fetuin-A should have CSF/serum quotients close to Q albumin or somewhat higher). Since the CSF/S quotients depend rather on hydrodynamic radius than on M.w., have the Authors any information about the hydrodynamic radius of fetuin-A compared to albumin? (partly discussed on p. 25 rows 516-523, but many Q fetuin-A values are much lower than corresponding Q Alb values also in patients with normal Q Alb).

We have studied the physical chemistry of fetuin-A and the calciprotein particles containing fetuin-A in detail. The distribution quotients may in fact be greatly affected by the state of fetuin-A at the time of sampling. Whether or not fetuin-A circulated as a monomer or as part of the colloidal calciprotein particles is not known. In addition, glycosylation and phosphorylation may both affect the hydrodynamic radius because they change the net charge of the fetuin-A molecule, albeit much less than agglomeration of fetuin-A into calciprotein particles. 

We previously determined the hydrodynamic radius of fetuin-A monomer (3.1±0.2 nm) and of calciprotein particles ( up to 150 nm).

 1. Rochette, C. N. et al. A Shielding Topology Stabilizes the Early Stage Protein–Mineral Complexes of Fetuin‐A and Calcium Phosphate: A Time‐Resolved Small‐Angle X‐ray Study. Chembiochem 10, 735–740 (2009).

 2. Wald, J. et al. Formation and stability kinetics of calcium phosphate –fetuin-A colloidal particles probed by time-resolved dynamic light scattering. Soft Matter 7, 2869–2874 (2011)

3. Heiss, A. & Schwahn, D. Handbook of Biomineralization: Biological Aspects and Structure Formation. 415–431 (2008) doi:10.1002/9783527619443.ch24.

Presently, we can only speculate on the aggregation state of fetuin-A in blood vs. CSF, but we will certainly come back to these questions once better methods for the detection of calciprotein particles in blood and CSF have been developed. This is the topic of ongoing research in our and other labs. 

2) the Authors tried to perform some kind of quantitation of the phosphorylated fetuin-A; what about the precision of these data (%CV in repeated experiments or from the literature etc.), and do the Authors think such quantitative attempts are possible and worthwhile for their glycosylation studies?

We completely agree that deglycosylation or phostag-PAGE in combination with immunoblotting are semi-quantitative at best. Western Blotting simply is NOT strictly quantitative. However, the method was suitable for this pilot study. We did not have the means and funds to perform quantitative protein-MS on the relatively large number of samples. As one of few laboratories in the world we have decades of experience with fetuin specific antibodies, both homemade and commercial. We prefer homemade antibodies, because in our hands they invariably outperform commercial products in terms of specificity and sensitivity.

3) The Authors used enzyme digestion to study post-translational modifications of fetuin-A. It would be interesting if they wish at least to mention the use of mass spectrometry for such a purpose and why they did not choose this analytical tool in their study (not suitable? not available? other reason?)

In collaborative work, we performed protein-MS many times. Like blotting, conventional protein-MS counting percentage of coverage etc. is also semi-quantitative at best. Truly quantitative protein phosphorylation analysis by MS requires labeling and standard peptides to establish recovery and sensitivity for the detection of each phospho/vs dephospho peptides in a given MS instrument (Kusebauch, U. et al. Human SRMAtlas: A Resource of Targeted Assays to Quantify the Complete Human Proteome. Cell 166, 766 778 (2016). Senior author WJ-D personally witnessed this first-class work during his Sabbatical 2015 at the Systems Biology Institute in Seattle/Washington. Truly quantitative protein-MS work cannot presently be performed in our clinical routine lab, let alone in our own lab.

Minor Criticisms:

Abstract, p.2, row 23: MS - abbreviation not explained (multiple sclerosis?), I would recommend to drop it from the Abstract since the role of fetuin-A as a biomarker for MS is far from being firmly established. 

We agree. The abstract now states "under investigation as a biomarker". Now row 26.

p.4 row 52 and p.22 row 449: Greek symbols or letters should be used consistenly for TNF-alpha and IL-1beta throughout the text

done

p.10 row 188 - "multiple" instead of "multipe" 

done, now row 239 

p.10 - subscript should be used appropriately in the formulas of chemical compounds (e.g. for hydrogen peroxide)

done e.g. row 233/234

p. 11 row 192: "sodium dodecyl sulfate" instead of "sodiumdodecylsulfat"; row 208: "polyacrylamide" instead of "polyacrylamid"

done now row 244 and 262

p.13 row 237: "2+" should be in superscript

done, row 290

Thank you so much for pointing out these typos!

p.13 row. 246 please provide brief description of the in-house AS5359 anti-human fetuin-A antibody, or provide an appropriate reference in the Reference list

we added reference 59, which describes the use of this polyclonal antiserum for immunoblotting. Now row 299

p. 16 rows 307-308: it is not clear what the Authors wished to say, and it seems to be related to Fig. 1F rather than Fig. 1G; please try to formulate in a clearer way

p. 16 row 317: Fig. 1G rather than 1H; row 321: Fig. 1H rather than 1G

Indeed, there was a mixup of the figures. This was corrected. Now row 366

p. 17 rows 339-343: this should be stated in the Methods section (see above) 

done as suggested, see above

p.19 row 392: incorrect unit for the range of CSF phospho fetuin-A concentrations (mg/ml instead of micrograms/ml)

corrected row 453, Thank you for spotting this error.

p. 22 row 444: please use "CSF" instead of "liquor"

done row 509

p. 26 row 533: "in the context" rather than "in the contexts"

done row 602

Tables S6 and S7 have identical Table title; I would recommend either to use different Table titles or to merge in one Table

Done

Once again we thank both reviewers for their meticulous review of our manuscript.

---

## [Decision Letter · Decision Letter 1]

26 Sep 2022

Post-translational modifications glycosylation and phosphorylation of the major hepatic plasma protein fetuin-A are associated with CNS inflammation in children.

PONE-D-22-12868R1

Dear Dr. Jahnen-Dechent,

We’re pleased to inform you that your manuscript has been judged scientifically suitable for publication and will be formally accepted for publication once it meets all outstanding technical requirements.

Kind regards,

Pavel Strnad

Academic Editor

PLOS ONE

Additional Editor Comments (optional): Thank you for submitting this nice work to PLoS One!

Reviewers' comments:

Reviewer's Responses to Questions

**Comments to the Author**

1. If the authors have adequately addressed your comments raised in a previous round of review and you feel that this manuscript is now acceptable for publication, you may indicate that here to bypass the “Comments to the Author” section, enter your conflict of interest statement in the “Confidential to Editor” section, and submit your "Accept" recommendation.

Reviewer #1: All comments have been addressed

Reviewer #2: All comments have been addressed

2. Is the manuscript technically sound, and do the data support the conclusions?

Reviewer #1: Yes

Reviewer #2: Yes

3. Has the statistical analysis been performed appropriately and rigorously? 

Reviewer #1: Yes

Reviewer #2: Yes

4. Have the authors made all data underlying the findings in their manuscript fully available?

Reviewer #1: Yes

Reviewer #2: Yes

5. Is the manuscript presented in an intelligible fashion and written in standard English?

Reviewer #1: Yes

Reviewer #2: Yes

6. Review Comments to the Author

Reviewer #1: I congratulate the authors on their excellent research in a very complex field of neurology. Hopefully this work eventually leads to translational diagnostic and prognostic biomarkers in the management of CNS inflammatory conditions.

There is a great clinical need for acute markers of neuro-inflammation in both paediatric and adult patients that can diagnose inflammatory conditions at a very early stage of disease progression. This would assist neurologists in the differential diagnoses of inflammation due to autoimmunity, infection, cancers and anatomical causes as well as in following the response to treatment and the course of the disease.

Existing inflammatory biomarkers such as CSF Neopterin and Cytokines/Chemokines are highly sensitive for neuro-inflammation and they correlate very well with the state of inflammation following treatment with immunotherapy; but unfortunately they are nonspecific markers of inflammation. The hunt for a differential biomarker of CNS inflammation continues.

Due to the acute nature of neuroinflammation, it is also vital for the specialist laboratory to be able to test and report these biomarkers within a day or two to limit the damage that ensues CNS inflammation. This would require assays which can be fully automated, are random access and fit for clinical purposes.

Wish you the best for your future research.

Reviewer #2: The Authors did a tremendous amount of work, and although their manuscript is long and not easy to follow, I believe it paves new and promising ways to analyse not only fetuin A but possibly also other CSF proteins.

I appreciate exhausting answers to the Rewievers´ questions and meticulous revision of the Manuscript.

I would only recommend to check/correct the following issues in Tables 1 and 5: Serum protein is given in g/l and not g/dl. Also, "Blood-CSF barrier (BCB) disorder" might be preferable to the term "Blood-CSF border ..." in these Tables. Finally, on Page 11, row 234, there is missing "d" in the word "sialidase-Au". In a legend to Fig. 1(G) (page 16, row 382), "scatter plot" should be used instead of "scatter blot".

The Reviewer would like to take this opportunity to wish the Authors lasting enthusiasm and many success in their further research.

7. PLOS authors have the option to publish the peer review history of their article (what does this mean?). If published, this will include your full peer review and any attached files.

Reviewer #1: **Yes: **Sushil Bandodkar

Reviewer #2: No

---

## [Editor Report · Acceptance letter]

29 Sep 2022

PONE-D-22-12868R1 

Post-translational modifications glycosylation and phosphorylation of the major hepatic plasma protein fetuin-A are associated with CNS inflammation in children 

Dear Dr. Jahnen-Dechent:

I'm pleased to inform you that your manuscript has been deemed suitable for publication in PLOS ONE. Congratulations! Your manuscript is now with our production department. 

Kind regards, 

on behalf of

Dr. Pavel Strnad 

Academic Editor

PLOS ONE